# Interplay between the DNA Damage Response and Immunotherapy Response in Cancer

**DOI:** 10.3390/ijms232113356

**Published:** 2022-11-01

**Authors:** Elizabeth Chun Yong Lee, Jessica Sook Ting Kok, Bin Tean Teh, Kah Suan Lim

**Affiliations:** 1Division of Medical Oncology, National Cancer Centre, Singapore 169610, Singapore; 2Laboratory of Cancer Epigenome, National Cancer Centre Singapore, Singapore 169610, Singapore; 3Cancer and Stem Cell Biology, Duke-NUS Medical School, Singapore 169857, Singapore

**Keywords:** DNA repair, cancer, genome instability, immunotherapy

## Abstract

Genome instability and immune evasion are both defining hallmarks of cancer. Tumorigenesis is frequently initiated when there is DNA damage to a proto-oncogene or tumor suppressor gene and DNA repair mechanisms are lost or insufficient to correct the damage; immune evasion then prevents the host immune system from recognizing these transformed cells. Therapies targeting genomic instability and immune evasion have been effectively used to treat cancer. Genotoxic therapies such as chemoradiation have been employed in cancer treatments for several decades, while immunotherapy is a relatively new class of cancer therapy that has led to disease regression even in patients with advanced cancer. Several recent studies have shown synergy between both classes of therapy targeting these two defining hallmarks of cancer, and different mechanisms are proposed to be involved. Here, we review the different classes of DNA damage, their links to cancer, and their contribution to immunotherapy responses, as well as the different models that are currently being used to study tumor–immune interactions.

## 1. Introduction

One of the major hallmarks of cancer is genome instability resulting from DNA damage caused by external insults and malfunctions in the cell’s DNA damage response (DDR) mechanisms [1,2]. Both external insults such as UV radiation and cigarette smoke and endogenous insults such as reactive oxygen species (ROS) can lead to DNA damage in proto-oncogenes or tumor suppressor genes and lead to the development of melanoma, lung, and other types of cancers [3,4,5]. When DNA damage occurs in a cell, cellular DDR gets activated to repair the damage that has been incurred. The type of DDR pathway employed is specific to the type of DNA damage, the location of the damage, and the cell cycle stage. For example, homologous recombination (HR) only occurs during the S, G2, or M phases, while NHEJ occurs throughout the cell cycle to repair DNA double-stranded breaks [6]. On the other hand, transcription-coupled nucleotide excision repair (TC-NER) only occurs during transcription and at transcribed genes, when RNA polymerase stalls at a DNA lesion, while global genomic nucleotide excision repair (GG-NER) happens throughout the genome [7,8,9]. 

Malfunctions in the DDR can either be caused by mutations in DDR genes or epigenetic alterations leading to the downregulation of DDR genes. For example, BRCA1, which is involved in HR, has been found to be mutated at both germline and somatic levels, or epigenetically silenced in cancer [10]. The loss of DDR pathway genes leads to unchecked mutations in the genome, which can result in tumorigenesis when these genomic alterations affect proto-oncogenes or tumor suppressor genes. Indeed, cancer cells frequently have DDR defects, leading to genomic instability. However, the loss of one DDR pathway can also lead to increased dependency on other pathways, resulting in targetable vulnerabilities. Mutations in the HR proteins *BRCA1* and *BRCA2* cause HR deficiency, replication fork instability, increased mutations, and genomic instability [11,12]. However, this HR deficiency and replication fork instability also lead to extreme sensitivity to PARP inhibitors, which has led to the development of a new class of drugs that leverage synthetic lethality in DNA repair [13,14,15].

Another promising class of cancer therapeutics that work well in cancers with certain classes of DDR defects are immune checkpoint inhibitors (ICIs) [16,17,18,19,20]. Immune checkpoint proteins such as PD-1 and CTLA4 regulate the immune system by binding to partner proteins on antigen-presenting cells, ensuring that the healthy host cells are not targeted by the host immune response [21,22]. ICIs work by blocking checkpoint proteins from binding to their respective targets, allowing the T cells to target the tumor cells [23]. Treatment with ICIs can lead to a durable response even in patients with advanced disease. However, only a small subset of treated patients responds to ICIs, underscoring an urgent need to identify reliable biomarkers for ICIs and methods to enhance ICI treatment in patients [23]. Tumors with DDR defects such as tumors with mismatch repair defects (MMRD) and POLE-mutant tumors (hypermutated) respond significantly better to ICI [16,17,19,20]. These hypermutated tumors harbor more neoantigens due to an increase in tumor mutational burden (TMB) and are therefore more immunogenic. Indeed, TMB is one of the most frequently used biomarkers for response to immunotherapy. 

Genotoxic cancer treatments also synergize well with ICIs. One commonly employed genotoxic treatment is radiation therapy (RT), which induces cancer cell death through ionizing radiation. A large clinical trial, the PACIFIC trial, found that patients with non-small-cell lung cancer had significantly longer progression-free survival when treated with a combination of chemoradiation and a PD-(L)1 inhibitor [24]. Indeed, the therapeutic efficacy of RT itself is closely linked to host immunity. A demonstration of the link between RT and host immunity is the abscopal effect, which we will discuss in detail in this review [25,26,27,28]. Interestingly, irradiation of tumor cells has also been found to upregulate PD-L1 both in vitro and in vivo, suggesting that RT sensitizes tumor cells to PD-(L)1 inhibitor treatment potentially through increased expression of its target protein(s) [29,30,31,32,33]. 

Chemotherapy, another common form of genotoxic treatment, uses a class of pharmaceutical agents that cause DNA damage through several different mechanisms, such as DNA alkylation and crosslinking, anti-metabolites that inhibit RNA or DNA synthesis, and topoisomerase inhibition [34]. Chemotherapy is frequently given in conjunction with RT or surgery. Both chemotherapy and RT have been shown to modulate the host immune response and synergize with ICIs [35,36,37,38]. 

In this review, we will discuss the different types of DNA damage and repair pathways, the association between genomic instability and tumorigenesis, the link between DNA damage and immunotherapy response, and the various models that are being employed to study these tumor–immune interactions.

## 2. The Different DNA Damage Repair Pathways and Their Associations with Cancer

### 2.1. Double-Stranded Breaks—NHEJ, HR, MMEJ, SSA

Double-stranded breaks (DSBs) are a deleterious form of DNA damage that results in cell cycle arrest, apoptosis, cellular senescence, or carcinogenesis [39,40]. ATM is activated and autophosphorylated in response to DSBs, leading to cell cycle arrest mediated by Chk2 [41,42,43,44,45]. DSBs are repaired via HR, non-homologous end joining (NHEJ), microhomology-mediated end joining (MMEJ), or single-strand annealing (SSA) [46,47,48].

HR has the highest fidelity [49]. Damaged DNA is processed by the MRN complex and CtIP via end resection, allowing RPA to bind to the single-stranded overhang [2]. RAD51 is recruited by the BRCA1, BRCA2, and PALB2 complex, displacing RPA. Processed DNA filament binds to the undamaged DNA via strand invasion, synthesizing the new complementary DNA strand [2]. Germline mutations in HR proteins can increase the risk of breast, ovarian, prostate, and pancreatic cancers [10,50]. PARP inhibitors and USP1 inhibitors are synthetically lethal with these tumors and have the potential to be pharmacologically exploited [13,51]. Germline biallelic mutations in BRCA1/2 results in the hereditary syndrome Fanconi anemia, which also carries an increased risk of early-onset hematologic malignancies [52,53,54]. 

NHEJ is relatively accurate and activated by the direct ligation of the broken strands of DNA via the end-binding proteins Ku70-Ku80 and 53BP1. RIF1 prevents DNA from being resected, while recruiting DNA-PKcs for end processing. XRCC4 and DNA ligase IV are also recruited to the DSB to ligate the DNA ends and fill in any gaps in the sequence [55]. Germline mutations in NHEJ proteins are characterized by immunodeficiency as well as radiosensitivity [56,57]. In cancers, reduced expression or loss of some NHEJ proteins including 53BP1 and Rev7 can lead to PARP inhibitor resistance in BRCA1-deficient cancers [58,59,60,61].

MMEJ, also known as Alt-EJ, is less accurate than NHEJ, giving rise to a large number of chromosomal translocations [62]. It is initiated by end resection [63]. PARP-1 binds to the DSB ends, recruiting MRN, and is followed by end bridging. ERCC1/XPF and FEN1 remove the 3′ tails and FEN1 removes the 5′ tails. POLQ then fills the gaps via DNA synthesis, and DNA is ligated by LIG3/XRCC1 [63]. This pathway is thought to play a greater role in HR-deficient cancers as levels of POLQ are increased in the absence of HR [64,65].

SSA is a less well-studied form of DSB repair that involves Rad50, Rad52, and RPA, and is a relatively low-fidelity pathway, resulting in deletions of repetitive sequences [66]. The inhibition of Rad52 is synthetically lethal in HR-deficient cells, suggesting that SSA might be upregulated in HR-deficient cancers [67].

### 2.2. Single-Stranded Breaks—NER, BER, MMR

Single-stranded breaks (SSBs), base damages, and DNA mismatches are repaired by nucleotide excision repair (NER), base excision repair (BER), or DNA mismatch repair (MMR) [9,68,69]. When SSBs are not repaired properly, the more deleterious DSB could occur.

Bulky lesions such as pyrimidine dimers are repaired via global genomic NER (GG-NER) or transcription-coupled NER (TC-NER). GG-NER repairs damage detected by XPC-Rad23B or DDB1-DDB2 [9]. TC-NER repairs DNA lesions on transcribed DNA by recruiting Cockayne syndrome proteins CSA and CSB [9,70]. Dual incision and repair synthesis occurs when ERCC1-XPF binds to the TFIIH complex [9]. Biallelic germline mutations in GG-NER pathway proteins cause xeroderma pigmentosa (XP), a UV sensitivity syndrome which increases the risk of skin cancers [71,72]. Interestingly, while germline TC-NER pathway mutations are not associated with cancer, somatic TC-NER pathway mutations have been identified in a subset of epithelial ovarian cancers with increased sensitivity to platinum-based chemotherapy and PARP inhibitors [73,74].

Base damages caused by oxidation, deamination, and alkylation are repaired via short-patch (SP)-BER or long-patch (LP)-BER [68,75]. AP endonucleases make a strand incision, allowing dRPase to remove the affected nucleotide. SP-BER is facilitated by XRCC1 and pol β protein, and LP-BER is facilitated by XRCC1, PARP1/2, and FEN1. Polynucleotide kinase-phosphatase (PNKP) removes the phosphates from the 3′ end and phosphorylates the 5′ end. DNA synthesis then occurs, and DNA ligase ligates the DNA ends together [68,76,77]. Biallelic germline mutation of the BER glycosylase MUTYH causes higher risk of MUTYH-associated polyposis (MAP), predisposing carriers to colorectal cancers [78,79]. Additionally, POLB is mutated in up to 40% of colorectal tumors, and polymorphisms of other BER genes carry an increased risk of lung, gastric, and esophageal cancers [80,81,82].

DNA mismatch repair (MMR) repairs base–base mismatches and insertion–deletion mutations that occur during DNA replication and recombination [69,83]. MSH2-MSH6 detects smaller stretches of mismatches while MSH2-MSH3 detects longer stretches. MLH1-PMS2, PMS2, and EXO1 excise the mismatched DNA. DNA synthesis is initiated, and DNA ligase finishes the repair [69]. Germline MMR gene mutations lead to microsatellite instability (MSI) and are associated with an increased risk of colorectal and endometrial cancer [84,85,86,87,88]. Somatic mutations or epigenetic silencing of MMR pathway components also result in MSI in colorectal cancers [87]. Homozygous germline mutations result in constitutional mismatch repair-deficiency cancer syndrome (CMMRD) which has a stronger cancer penetrance and earlier onset [89].

### 2.3. Fanconi Anemia

The *Fanconi anemia* (FA) pathway repairs DNA interstrand crosslinks (ICL) [90]. ICLs affect both strands of the DNA double helix and result in DNA breaks and chromosomal rearrangements. The FAAP24-FANCM-MHF1-MHF2 complex detects ICLs, recruiting the FA core complex to the chromatin. Monoubiquitination of FANCD2-I heterodimers positions the complex and recruits downstream repair factors. MUS81, SLX4, and FANCQ cleave the crosslinked DNA via unhooking. The crosslinked nucleotide base paired to the complementary DNA and the lesion is bypassed via translesion synthesis [91,92,93]. FANCD2-I proteins are then deubiquitinated by the USP1-UAF1 complex [91,92,93,94,95,96]. Mutations in the FA pathway proteins lead to *Fanconi anemia*, associated with leukemias, ovarian, breast and other cancers [92].

### 2.4. R-Loops

R-loops form when an RNA strand displaces the non-template DNA strand behind RNA polymerase during transcription, creating a triple-stranded nucleic acid structure [97,98,99]. Physiologic R-loops are scheduled R-loops that form during transcription, whereas pathologic R-loops are unscheduled and disrupt cellular processes, increasing genomic instability [97,100,101]. R-loops are resolved by nucleases and RNA-specific helicases [98,101,102,103]. SETX unwinds the DNA–RNA hybrid while RNaseH enzymes degrade the RNA strand, allowing the complementary DNA strands to reanneal to form the double helix [101,104,105,106,107,108].

### 2.5. Translesion Synthesis

The translesion DNA synthesis (TLS) pathway gets activated when bulky DNA lesions block replicative DNA polymerase progression. PCNA gets monoubiquitinated and switches to bind a TLS polymerase which fills in the gap with low fidelity [109]. This pathway is more permissive, low fidelity, and prone to errors, but it allows cells to bypass a lesion that might otherwise lead to a collapsed replication fork and the formation of deleterious DSBs [109].

While mutations in the R-loop processing and TLS pathway components have not been specifically linked to cancer predisposition, both contribute to replication stress and genomic instability and thus are likely to contribute to tumorigenesis. A summary of the different types of DNA damage, modes of DNA repair, examples of genes involved, and examples of related cancers are given in Table 1 and Figure 1.

## 3. Link between DDR Pathway Mutations and the Response to Immunotherapy

Cancer immunotherapy is a promising new class of therapy that harnesses the patient’s immune system to kill cancer cells [23]. Immune checkpoint blockade is a form of immunotherapy that prevents inhibitory immune checkpoints from being activated, allowing the immune response to target cancer cells [23]. PD-1, or programmed cell death-1, is an immune checkpoint that is expressed on CD4+ and CD8+ T cells [110,111]. Its function is to prevent an autoimmune response by regulating T cell proliferation and IFNγ production to create an immunosuppressive microenvironment [110,111]. PD-L1 is its binding partner, expressed primarily on antigen-presenting cells such as dendritic cells and tumor cells. When PD-1 binds to PD-L1, T cell activation and proliferation is inhibited [111,112,113]. Clinical application of PD-1/PD-L1 inhibitors have been widely tested on a number of different cancer types such as melanoma, lung cancers, and colon cancers. Pembrolizumab, nivolumab, atezolizumab, and durvalumab are examples of PD-1/PD-L1 inhibitors that have been FDA approved for the treatment of cancers [114,115].

While cancer immunotherapy can result in tumor remission in some patients with advanced disease, the proportion of treated patients that respond is small [116]. It is therefore important to (1) identify better biomarkers of response and (2) identify ways to enhance the response to cancer immunotherapy [116]. Several studies have found DNA repair-deficient tumors to be extremely sensitive to immune checkpoint blockade. These tumors, which include MMRD and HRD tumors, are characterized by a high TMB, which is a good biomarker of response to PD-(L)1 blockade [16,17,20,117,118,119]. It was hypothesized that a higher TMB corresponds to an increase in immunogenic neoantigens, leading to an enhanced response to immune checkpoint blockade. HRD tumors also frequently have increased PD-L1 expression in the tumor microenvironment [120]. Indeed, the presence of DSBs (likely to be higher in HRD tumors) has been found to upregulate PD-L1 expression in cancer cells, in a process that requires ATM/ATR/Chk1 kinases and is driven by STAT1 and STAT3 signaling, providing a potential explanation for the elevated PD-L1 seen in HRD tumors [117]. Interestingly, elevated PD-L1 expression is also a well-established biomarker of response for anti-PD-(L)1 therapy [121]. Consistent with these observations, HRD was found to be a superior biomarker of response in a study conducted on patients enrolled in a phase I/II trial of niraparib and pembrolizumab in ovarian cancer [122]. More recently, loss-of-function mutations in the DNA binding or exonuclease domains on polymerase epsilon, associated with hypermutated tumors and increased TMB, have also been reported to respond well to anti-PD-(L)1 therapies [123].

While one might expect mutations in DNA repair genes to predict an increase in tumor mutational burden, this is not always true if the mutation occurs in a region that is not functionally important. A scoring system such as the HRD score, which combines information about mutations in DNA repair genes and genomic scars that suggests a HR defect, might therefore be important to determine the loss of DNA repair capacity [124]. The usage of DNA repair defects scoring systems as biomarkers of response to immunotherapy has not been explored extensively, but a recent small study in non-small-cell lung cancers demonstrated that usage of the HRD score was able to accurately predict response to immuno-neoadjuvant therapy [125]. A large retrospective bioinformatics analysis of pan-cancer TCGA data also correlated HRD score with an immune-sensitive phenotype in tumors [60]. It would therefore be interesting to determine the predictive value of such a scoring system as a biomarker for immunotherapy in future larger-scale studies.

The cellular role of PD-L1 outside of immunity has been under-studied. It was recently reported that PD-L1 binds to the transcripts of several DNA repair proteins, playing an important role in regulating DNA repair [126]. This novel DNA repair role confers radiation sensitivity to cells when PD-L1 is silenced. Another study found that PD-L1 plays an important role in homologous recombination by promoting BRCA1 foci formation. Consequently, PD-L1 knockout led to increased PARP inhibitor sensitivity in mouse tumors [127]. These studies uncovered previously unknown cell-intrinsic roles of PD-L1 and provide a potential explanation for the upregulation of PD-L1 in response to DNA damage—to perform DNA repair. Additionally, another recent study found that PD-L1 translocates into the nucleus to regulate immune gene expression [128]. This raises the interesting possibility that PD-L1 might play a more direct role in DNA repair, owing to its ability to bind to DNA. Crosstalk between the DNA repair role of PD-L1 and its role in immunity likely exist and remain to be investigated.

While deficiencies in HR and MMR pathways have both been demonstrated to lead to improved response with cancer immunotherapy, deficiencies in several other key DNA repair pathways, including TLS, FA, R-loop processing pathways, and ATM mutations in tumors, have all been under-studied in the context of cancer immunotherapy response. Since a high TMB is frequently observed in cancers that have DDR defects and high TMB is a good prognostic marker for immunotherapy response, it is very likely that these other DDR-deficient cancers also respond well to immune checkpoint blockade and should be explored further in future studies.

## 4. Link between DNA Repair, RNA Editing, R-Loops, and the Response to Immunotherapy

Altered RNA metabolism and processing frequently lead to the accumulation of R-loops, increased DNA damage, and tumorigenesis [129,130,131]. In particular, repetitive sequences have been found to be prone to aberrant R-loop accumulation [132,133]. Treatment of cancer cells with increased repetitive sequences with a reverse transcriptase inhibitor 3T3 led to an increase in RNA–DNA hybrids, type I interferon response, and DNA damage response, and might sensitize tumors to immune checkpoint blockade [134].

The RNA-editing enzyme ADAR1 has also been found to be important in response to cancer immunotherapy. Ablation of ADAR1 in tumor cells inflamed tumors and sensitized tumors to immune checkpoint blockade due to the increased innate immune response recognition of dsRNA upon decreased A-to-I RNA editing [135]. Interestingly, ADAR1 also prevents R-loop accumulation, and it is possible that pathogenic R-loop accumulation and DNA damage resulting from the loss of ADAR1 activates the innate immune response as well [136]. It is worth mentioning that ADAR1 had also been found to be important in preventing Z-RNA accumulation, and this leads to resistance to immune checkpoint blockade through a different mechanism that is beyond the scope of this review [137].

## 5. Link between Genotoxic Cancer Therapies and the Response to Immunotherapy

Besides DDR deficiencies, genotoxic treatments can also improve response to immune checkpoint blockade in cancer. Both chemotherapy and radiotherapy have been shown to synergize with immune checkpoint blockade, and several clinical trials are ongoing for the co-treatment of tumors with PARP inhibitors and immune checkpoint inhibitors [38,138,139]. 

Radiotherapy in particular exhibits remarkable synergy with the host immune response. The abscopal effect, a rare but interesting phenomenon where the radiation treatment of one tumor site leads to the regression of tumors at distant metastatic sites in the same patient, provided early evidence that tumor irradiation synergizes with host immunity to achieve systemic anti-tumor effects [27]. In mice with fibrosarcoma lacking T cells, it was also observed that a significantly higher dose of radiation was required to achieve the same anti-tumor effect, suggesting that T cells were important in the anti-tumor activity of radiation [140]. More recently, Lee et al. and Gupta et al. also showed that tumor response to high-dose ablative radiation in mice is dependent on both CD8+ cytotoxic T cells and dendritic cells, further demonstrating the synergistic relationship between radiotherapy and host immunity [141,142]. Immune checkpoint inhibitors enhance this synergistic relationship, resulting in further improved response in patients [37].

One potential mechanism that might explain this synergy is the increase in neoantigens in tumors after treatment with genotoxic therapies. In support of this mechanism, a pivotal study in lung cancer demonstrated that increased sensitivity to anti-CTLA4 immunotherapy in irradiated tumors can be directly attributed to the resulting upregulation of neoantigen presentation. Using both whole exome sequencing and RNA sequencing to detect somatic mutations in tumors from a patient with complete response to a combination of radiation and anti-CTLA4 therapy, the authors demonstrated that irradiation induced a rapid expansion of T cells that targets a neoantigen which was upregulated post-radiation [143].

Another mechanism of synergy between genotoxic cancer therapies and immunotherapy is activation of the cyclic GMP–AMP synthase (cGAS)-stimulator of interferon genes (STING) pathway. Upon genotoxic treatment of cancer cells, nuclear integrity is often compromised, leading to the leakage of DNA into the cytosol [144,145,146]. The cGAS-STING pathway is an innate immune sensor that triggers a cellular immune response upon the detection of cytosolic DNA [147,148]. cGAS binds to cytosolic DNA and produces cGAMP, which in turn activates STING [149,150,151,152,153]. Upon activation, STING triggers the phosphorylation of interferon regulatory factor 3 (IRF3) and nuclear factor-κB (NF-κB) inhibitor IκBα by TANK-binding kinase 1 (TBK1) and IκB kinase (IKK), respectively. These lead to the nuclear translocation of IRF3 and NF-κB, activating the transcription of cytokines, chemokines, and type I interferons (IFNs) [154,155,156]. These immunological agents are then secreted from the cells to facilitate the infiltration of anti-tumor CD4+ and CD8+ T cells into the tumor microenvironment, leading to enhanced anti-tumor immunity and response to immune checkpoint blockade [144,157,158]. Consequently, the development of pharmaceutical STING pathway agonists as a complementary therapy to cancer immune checkpoint blockade is a subject of intense research. 

One of the best-studied classes of STING activators is cyclic dinucleotides (CDN) which is formed by the catalysis of ATP and GTP [159]. cGAMP is an example of a CDN [149,150,151,152]. CDNs can be directly derived from bacteria. However, their hydrophilic nature, small size, and low stability, delivery, and bioavailability have proven to be a challenge. For example, CDNs are readily degraded by enzymes such as ENPP1. To overcome this, next-generation STING agonists include inhibitors of CDN degraders including ENPP1, complementing cancer immunotherapy [160,161,162,163]. 

ENPP1 is a type II transmembrane glycoprotein that degrades the STING ligand [160]. A recent study has shown that high expression of ENPP1 can be correlated to poorer prognosis in high-grade serous ovarian carcinoma, suggesting that it can be used as a prognostic biomarker [164]. The knockdown of ENPP1 has also shown to reduce growth and increase cell death in this study. In another recent study, treatment with a pan-tyrosine kinase inhibitor pazopanib led to an immunogenic anti-tumor response, where PD-L1 expression and STING pathway were both upregulated. Interestingly, while ENPP1 was highly expressed in tumor tissue before treatment, expression decreased significantly after treatment with pazopanib, in agreement with STING pathway and PD-L1 upregulation [165].

Besides ENPP1, TREX1 is an exonuclease that also acts to antagonize the cGAS-STING pathway in cells, likely through the degradation of cytosolic DNA [166,167,168,169]. Mutations in TREX1 lead to the neurological disorder Aicardi–Goutières syndrome, which is driven by constitutive activation of type I interferon signaling [170]. Upregulation of TREX1 has been shown to lead to resistance against radiotherapy-induced immunogenicity and chemotherapy through the downregulation of the cGAS-STING pathway [171,172,173]. Inhibition of TREX1 is therefore an attractive strategy to enhance immunotherapeutic responses [174,175].

Several pre-clinical and clinical genotoxic therapies have also been shown to upregulate the cGAS-STING pathway and potentiate cancer immunotherapy in animal models. ATM inhibition, ATR inhibition, Chk1 inhibition, PARP inhibition, and WEE1 inhibition have all been shown to lead to cGAS-STING pathway activation and potentiation of cancer immunotherapy [176,177,178,179,180,181]. 

## 6. Models to Study These Interactions

In order to study tumor immunology, we need reliable models that faithfully recapitulate the interaction between cells from the immune system and tumor cells. Current commonly used models include in vitro cell line co-culture models, genetically engineered mouse models (GEMM), and syngeneic or humanized animal models. 

Cell line co-culture models involve co-culturing T cells (either donor-derived or T-cell lines) with cancer cells. The T cells are typically artificially activated before the co-culture and might first be primed with an artificial antigen such as ovalbumin that is also expressed on co-cultured cancer cells. However, this model consists only of two different cell types and does not take into account other microenvironmental factors [182].

Syngeneic mouse models involve implanting tumor cell lines obtained from commonly used strains of mice into other mice of the same strain [183]. This model accurately recapitulates autologous immune interaction with tumors and microenvironmental factors, and tumors develop in a relatively short amount of time. However, the tumors that develop from this model typically lack the characteristic heterogeneity of cancers and are limited in scope for drug testing due to structural differences between human proteins and mouse proteins [184].

Humanized mice models overcome this problem by engineering human proteins to be expressed by mice or replacing the immune-compromised mice hematopoietic system with human hematopoietic stem cells (HSCs) or peripheral blood mononuclear cells (PBMCs). Using such a model, it is possible to construct mice “avatars” of a patient to study immunotherapy response by using cells from the patient’s tumor to inoculate NSG mice and transplanting CD34+ HSCs from the same patient into the mice [185,186]. The limitation of such an “avatar” model is the fact that patient tumor cells do not necessarily form tumors in mice, and the length of time it takes for patient HSCs to engraft and reconstitute the mouse immune system might be relatively long [187]. One way to overcome the long engraftment and reconstitution period would be to use PBMCs instead of HSCs. However, the use of PBMCs leads to graft-versus-host disease development rapidly (as short as a month) after bone marrow reconstitution and narrows the window for experimental design [188]. In general, humanized mice models are also prohibitively expensive and large studies using such models are rare.

GEMMs are mice models of cancers that have either oncogenic or tumor suppressor mutations engineered into their genomes to increase their chances of developing cancers [189,190]. These tumor models typically recapitulate the natural heterogeneity and complexity associated with cancer due to the process of immune editing as the tumor develops de novo. The increase in tumor heterogeneity also increases the neoantigen load. However, tumors take a long time to form in these models and immunotherapy treatment sometimes gives rise to inconsistent results due to the increase in heterogeneity and immune resistance of these tumors. These tumors are also limited in scope for drug testing due to structural differences between human proteins and mouse proteins [184].

Tumor-on-chip models are a form of indirect co-culture system that cultures tumor cells in the form of 3D tissue spheroids or fragments on a chip in a controlled environment [191]. The chip replicates the capillary system and tumor microenvironment by allowing a constant supply of nutrients while removing waste from the medium through a microfluidics system. The system also allows for more accurate observations of how both the tumor and surrounding cells react to different doses of drug treatments through serial sampling. However, tumor-on-chip models require specialized designs and incubators, and for these reasons they are less easily accessible compared to other models [191]. 

Patient-derived organoid tumor models are 3D ex vivo models derived from tumor tissues that better recapitulate intra-tumoral heterogeneity and cell–cell interactions in vivo [192,193]. One of the limitations to organoid systems for cancer research is the lack of immune components. As organoid models only retain the stromal components for extended culture, autologous immune components would need to be added separately in order to better mimic the tumor microenvironment. This can be done in the form of a 3D co-culture model [182,194]. Direct co-culture models involve the direct interaction between the cell types via their cell surface receptors, while indirect co-culture models involve either the physical separation of the cell types via a transwell or microfluidic system while sharing the growth medium, or by collecting and using the conditioned medium of one cell type to culture the other [182,191,194]. Differences in culture medium requirements for different cell types limit the number of cell types that can be co-cultured and the length of time each co-culture can be carried out. Currently, most direct co-culture experiments involving tumor-immune cells are short term. With the development of improved media formulations, longer-term co-cultures might become possible [182].

## 7. Discussion

Cancer is a disease that is defined by genomic instability and immune evasion. The advent of cancer immunotherapy has improved outcomes even for cancer patients with advanced disease. However, due to low rates of response and response heterogeneity, methods of enhancing cancer immunotherapy and improved biomarkers of response are urgently needed. In this review, we have discussed the various DNA repair processes that are frequently disrupted in cancer, giving rise to this genomic instability. We have also discussed how these DNA repair defects could contribute to sensitivity to traditional genotoxic therapies or cancer immunotherapy. We summarized the findings of publications that described how genotoxic treatments enhance cancer immunotherapy and DNA-repair-related biomarkers of response and also discussed the different model systems that are being used in immunotherapy studies. 

Going forward, one of the key areas that remains to be addressed includes the contribution of tumor heterogeneity to cancer immunotherapy response and how these might interact with genotoxic therapies. The intrinsic heterogeneity of tumors, with different cell populations consisting of different mixtures of mutations and microenvironmental changes, affect therapeutic responses in different ways. For example, hypoxic regions of the tumor are known to be resistant to radiation treatment. Resistance to radiation in hypoxic regions might thus inhibit synergy with cancer immunotherapy, leading to the survival and outgrowth of therapy-resistant populations of cells and tumor recurrence. Other microenvironmental and mutational heterogeneity might also influence the synergy between genotoxic therapies and immunotherapy; this is therefore an area that warrants further investigation.

The cellular DDR machinery is necessarily complex in order to coordinate and repair the wide-ranging forms of genotoxic insults that affect genomic DNA. Two of the major DDR pathways that remain to be explored comprehensively in the context of cancer immunotherapy co-treatment include the Fanconi anemia pathway and the R-loop resolution pathway. Targeted inhibitors to these pathways might enhance response to cancer immunotherapy in certain contexts, and tumors deficient in these pathways might also benefit significantly from immunotherapy treatments.

New and more targeted forms of radiation therapies such as proton therapy and carbon ion therapy allow higher doses of radiation to be used with fewer side effects in cancer. These are primarily used for tumors in deep-seated locations or locations that are difficult to target with conventional photon-based radiation due to their proximity to important organs. Whether this higher precision and dose translates to enhanced synergy with cancer immunotherapy also warrants further investigation. 

The fields of traditional genotoxic therapies and promising new cancer immunotherapy are merging due to their ability to synergize and enhance patients’ response to treatment. Here, we have provided a snapshot of the various DNA repair pathways that have been shown to synergize with or have the potential to synergize with cancer immunotherapy. While some mechanistic aspects of this synergy have been elucidated, much work remains to be done to expand this increased efficacy to a larger patient population. Since genomic instability is an inherent characteristic of cancer, we believe that identifying the appropriate targetable DDR pathway in each tumor, when performed in conjunction with immunotherapy, has the potential to benefit almost all cancer patients.

## Figures and Tables

**Figure 1 ijms-23-13356-f001:**
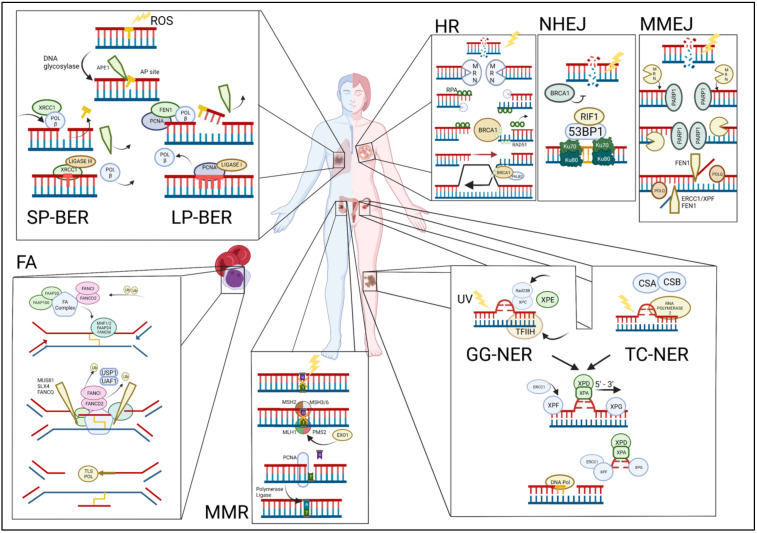
Overview of the DNA damage repair pathways and their commonly associated cancers (clockwise). TOP LEFT: Single-stranded breaks are repaired via either short-patch base excision repair (SP-BER) or long-patch BER (LP-BER). BER begins when the DNA base is damaged and is excised by the enzyme apurinic/apyrimidinic endodeoxyribonuclease 1 (APE1) and further processed by either SP-BER or LP-BER. Polymorphisms in BER pathway genes have been linked to an increased risk for the development of lung and other cancers. TOP RIGHT: Double-stranded breaks (DSBs) are repaired via homologous recombination (HR), non-homologous end joining (NHEJ), or microhomology-mediated end joining (MMEJ). HR: the MRE11-RAD50-NBS1 (MRN) complex detects double-stranded breaks. NHEJ: the Ku70/80 heterodimer detects and binds to the DSBs. MMEJ: the protein-coding gene poly (ADP-Ribose) polymerase 1 (PARP1) detects DSB mutations in HR pathway genes that are frequently associated with breast and other cancers. BOTTOM RIGHT: Nucleotide excision repair (NER) pathway is activated upon the detection of bulky DNA lesions and starts off as either a global genome NER (GG-NER) or a transcription-coupled NER (TC-NER). Mutations in TC-NER proteins have been identified in a subset of ovarian cancers, conferring increased sensitivity to platinum-based chemotherapy. BOTTOM MIDDLE: Mismatched repair (MMR) pathway is initiated by recruitment and binding of the MutLα complex on the damaged site. Mutations in MMR pathway genes are associated with colorectal and other cancers. BOTTOM LEFT: In the *Fanconi anemia* (FA) pathway, FANCM is first recruited to the site of ICL damage at the stalled replication fork along with FAAP24 and the heterodimer MHF1/2, followed by HR pathway mutations in FA pathway genes that lead to the *Fanconi anemia* syndrome characterized by a high incidence of childhood leukemias.

**Table 1 ijms-23-13356-t001:** Summary of Types of DNA damage, Modes of repair, Genes involved and related cancers.

Type of DNA Damage	Mode of DNA Damage Repair	Examples of Genes Involved	Examples of Related Cancers
Double-stranded breaks (DSBs)	Homologous repair (HR)	BRCA1/BRCA2	Breast, ovarian, prostate, pancreatic, and other cancers
	Non-homologous end joining (NHEJ)	53BP1, PTEN	General (contributes to chromosomal translocations)
Single-stranded breaks (SSBs)	Nuclear excision repair (NER) via GG-NER or TC-NER	XPC, XPE (UV-DDB2)	Melanoma and other skin cancers
	Base excision repair (BER)	MUTYH, XRCC1	Colorectal and other cancers
	DNA mismatch repair (MMR)	MSH2, MLH1	Colorectal, endometrial, and other cancers
Intra- and Interstrand crosslinks (ICLs)	*Fanconi anemia*, translesion synthesis (TLS)	FANCD1, FANCR	Myeloid leukemias and ovarian and breast cancers

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
