# Peer review of "Interplay between the DNA Damage Response and Immunotherapy Response in Cancer"

_ijms, 2022, doi:10.3390/ijms232113356_

Round 1

Reviewer 1 Report

This review by Lee et al. gives an overview of DNA repair pathways and a brief description of which of them are disrupted in particular cancers. It then goes on, more interestingly, to discuss the relationship between immunotherapy and DNA damage (responses). 

Unfortunately, I find much of the content very superficial and saying nothing that hasn't been said better and in much greater detail in numerous other publications. The discussion of the repair pathways is acceptable, at an undergraduate level, but adds nothing new or different. Most of the references are to reviews and very few to original research papers. For example, in the discussion of R-loops 3 reviews are listed and nothing more. It is, of course, difficult to say anything very original if one attempts to cover such a wide area in 12 pages. But even in this space there is an appreciable amount of repetition- for example, the extensive figure 1 legend largely repeats what has been said on pages 3 and 4. Similarly, parts of the introduction are repeated later in the manuscript; for example lines 391-4

 The abscopal effect, a rare but interesting phenomenon where the radiation treatment of one tumor site leads to the regression of tumors at distant metastatic sites in the same patient, provided early evidence that tumor irradiation synergizes with host immunity to achieve systemic anti-tumor effects[26]

are very similar to lines 68-70

 The abscopal effect, where RT at  one tumor location lead to the regression of tumors at distant metastatic sites, provided the first hint that tumor irradiation can synergize with the host immune system to achieve 70 anti-tumor effects[24-27]. 

The manuscript becomes more interesting when it goes on to consider the relationship between genotoxic therapies and immunotherapy. My recommendation would be to lose most of the material described on pages 3 to 8 and expand the cancer therapy/radiation/immunotherapy material also deleting the repetition.

There are a few factual errors in the manuscript.

For example, page 1 para 1, DNA damage can also be caused by endogenous factors such as ROS.

Line 99. The MRN complex does not phosphorylate ATM. ATM phosphorylation is generally considered to be autophosphorylation.

Line 124-the use of 'etc.' is hardly helpful. Readers would normally consult a review to obtain information they were unaware of. The use of 'etc.' would only be useful if you already knew the answer.

Pages 2-3. There is no mention of alt-NHEJ in this section-surely it is important.

Author Response

Reviewer 1:

This review by Lee et al. gives an overview of DNA repair pathways and a brief description of which of them are disrupted in particular cancers. It then goes on, more interestingly, to discuss the relationship between immunotherapy and DNA damage (responses). 

Unfortunately, I find much of the content very superficial and saying nothing that hasn't been said better and in much greater detail in numerous other publications. The discussion of the repair pathways is acceptable, at an undergraduate level, but adds nothing new or different. Most of the references are to reviews and very few to original research papers. For example, in the discussion of R-loops 3 reviews are listed and nothing more. It is, of course, difficult to say anything very original if one attempts to cover such a wide area in 12 pages. But even in this space there is an appreciable amount of repetition- for example, the extensive figure 1 legend largely repeats what has been said on pages 3 and 4. Similarly, parts of the introduction are repeated later in the manuscript; for example lines 391-4

Rebuttal:

We thank the reviewer for their comments. We have now included several original papers in the review, including several R-loop specific papers. We have also greatly condensed the repetitive segments.

The abscopal effect, a rare but interesting phenomenon where the radiation treatment of one tumor site leads to the regression of tumors at distant metastatic sites in the same patient, provided early evidence that tumor irradiation synergizes with host immunity to achieve systemic anti-tumor effects[26]

are very similar to lines 68-70

The abscopal effect, where RT at  one tumor location lead to the regression of tumors at distant metastatic sites, provided the first hint that tumor irradiation can synergize with the host immune system to achieve 70 anti-tumor effects[24-27]. 

Rebuttal:

We thank the reviewer for pointing out the repetitions. We have now removed the repetitive portions of the manuscript.

The manuscript becomes more interesting when it goes on to consider the relationship between genotoxic therapies and immunotherapy. My recommendation would be to lose most of the material described on pages 3 to 8 and expand the cancer therapy/radiation/immunotherapy material also deleting the repetition.

Rebuttal:

We thank the reviewer for the recommendations. We have now greatly condensed the first part of the manuscript and expanded on the later part of the manuscript, including detailed paragraphs about HRD scoring as a biomarker and the role of PD-L1 in DNA repair under the section “Link between DDR pathway mutations and the response to immunotherapy”, a detailed paragraph about TREX1 under the section “Link between genotoxic cancer therapies and the response to immunotherapy”, expanded the section “Models to study these interactions” with more information about humanized mice and a paragraph about GEMMs, and added a new section on “Link between DNA repair, RNA editing, R-loops and the response to immunotherapy”.

There are a few factual errors in the manuscript.

For example, page 1 para 1, DNA damage can also be caused by endogenous factors such as ROS.

Line 99. The MRN complex does not phosphorylate ATM. ATM phosphorylation is generally considered to be autophosphorylation.

Line 124-the use of 'etc.' is hardly helpful. Readers would normally consult a review to obtain information they were unaware of. The use of 'etc.' would only be useful if you already knew the answer.

Rebuttal:

We thank the reviewer for bringing our attention to the above. We have now edited the text to include ROS in page 1 paragraph 1. We have also amended the text under the section “i) Double-stranded breaks - NHEJ, HR, MMEJ” to “ATM is activated and autophosphorylated in response to DSB”. Finally, we have removed the “etc” within the text.

Pages 2-3. There is no mention of alt-NHEJ in this section-surely it is important.

We have include MMEJ (also known as Alt-EJ) under the section “i) Double-stranded breaks - NHEJ, HR, MMEJ, SSA”.

Reviewer 2 Report

In this manuscript, Lim and colleagues review the interplay between the DDR (DNA damage response) and cancer immunotherapy. Even though the text is well-written, I question the importance of this text vis a vis the already published literature, as many other recent reviews cover this same topic (see PMID: 30370352; PMID: 34970273; PMID: 32612833; PMID: 33888558; PMID: 34376827). This includes a recent publication at IJMS (PMID: 33888558). Therefore, I do not think the text should be published without major reorganization and change in focus.

An important issue of concern is the extensive review of the DNA damage repair pathways in the first section. This takes about half of the review and is pointless because these pathways have been reviewed in many other texts including textbooks.

The authors should consider focus on recent novelties in the field, as this is a fast-moving topic of research. In the section “Link between DDR and the response to immunotherapy”, the authors mainly discuss HDR and MMR-deficient tumors. The authors should consider exploring in detail the recent advances in one of these pathways (HDR or MMR). The section “models to study these* interactions” is interesting and could be better explored and extended. Furthermore, when describing the link between DDR and cGAS-STING pathway, the authors should consider citing the important reviews on this topic, which include: PMID: 29622565; PMID: 32283785; PMID: 34249949; PMID: 35265630.

Overall, in this current form, the review overlaps with topics of research already covered by several reviews and major modifications must be done to make it suitable for publication.

*Consider clarifying “these”.

Author Response

Reviewer 2:

In this manuscript, Lim and colleagues review the interplay between the DDR (DNA damage response) and cancer immunotherapy. Even though the text is well-written, I question the importance of this text vis a vis the already published literature, as many other recent reviews cover this same topic (see PMID: 30370352; PMID: 34970273; PMID: 32612833; PMID: 33888558; PMID: 34376827). This includes a recent publication at IJMS (PMID: 33888558). Therefore, I do not think the text should be published without major reorganization and change in focus.

An important issue of concern is the extensive review of the DNA damage repair pathways in the first section. This takes about half of the review and is pointless because these pathways have been reviewed in many other texts including textbooks.

Rebuttal:

We thank the reviewer for the recommendations. We have now condensed the first part of the manuscript, which contains overlaps with topics of research covered by other reviews.

The authors should consider focus on recent novelties in the field, as this is a fast-moving topic of research. In the section “Link between DDR and the response to immunotherapy”, the authors mainly discuss HDR and MMR-deficient tumors. The authors should consider exploring in detail the recent advances in one of these pathways (HDR or MMR). The section “models to study these* interactions” is interesting and could be better explored and extended. Furthermore, when describing the link between DDR and cGAS-STING pathway, the authors should consider citing the important reviews on this topic, which include: PMID: 29622565; PMID: 32283785; PMID: 34249949; PMID: 35265630.

Rebuttal:

We thank the reviewer for the kind suggestions. We have now included a detailed paragraph on HRD score and recent advances in its usage as a biomarker under the section “Link between DDR pathway mutations and the response to immunotherapy”. We have also included a detailed paragraph on recent advances on the role of PD-L1 in DNA repair under the section “Link between DDR pathway mutations and the response to immunotherapy”, a detailed paragraph about TREX1 under the section “Link between genotoxic cancer therapies and the response to immunotherapy”, expanded the section “Models to study these interactions” with more information about humanized mice and a paragraph about GEMMs, and added a new section on “Link between DNA repair, RNA editing, R-loops and the response to immunotherapy”. Finally, we have also included the reviews suggested in the fourth paragraph under the section “Link between genotoxic cancer therapies and the response to immunotherapy”.

Overall, in this current form, the review overlaps with topics of research already covered by several reviews and major modifications must be done to make it suitable for publication.

Round 2

Reviewer 1 Report

This is an appreciable improvement on the previous version and is now a useful review.

The changes made in the 'marked up' version  have led to some grammatical errors but the manuscript is otherwise fine.

Reviewer 2 Report

The authors addressed all the issues raised and the paper should be accepted for publication.